# Genetics of Acromegaly and Gigantism

**DOI:** 10.3390/jcm10071377

**Published:** 2021-03-29

**Authors:** Anna Bogusławska, Márta Korbonits

**Affiliations:** 1Department of Endocrinology, Jagiellonian University Medical College, 31-008 Cracow, Poland; boguslawskaania@gmail.com; 2Centre for Endocrinology, William Harvey Research Institute, Barts and the London School of Medicine and Dentistry, Queen Mary University of London, London EC1M 6BQ, UK

**Keywords:** acromegaly, AIP, gigantism, FIPA, MEN1, somatotroph adenoma, pituitary neuroendocrine tumour, X-linked acrogigantism

## Abstract

Growth hormone (GH)-secreting pituitary tumours represent the most genetically determined pituitary tumour type. This is true both for germline and somatic mutations. Germline mutations occur in several known genes (*AIP*, *PRKAR1A*, *GPR101*, *GNAS*, *MEN1*, *CDKN1B*, *SDHx*, *MAX*) as well as familial cases with currently unknown genes, while somatic mutations in *GNAS* are present in up to 40% of tumours. If the disease starts before the fusion of the epiphysis, then accelerated growth and increased final height, or gigantism, can develop, where a genetic background can be identified in half of the cases. Hereditary GH-secreting pituitary adenoma (PA) can manifest as isolated tumours, familial isolated pituitary adenoma (FIPA) including cases with *AIP* mutations or *GPR101* duplications (X-linked acrogigantism, XLAG) or can be a part of systemic diseases like multiple endocrine neoplasia type 1 or type 4, McCune–Albright syndrome, Carney complex or phaeochromocytoma/paraganglioma-pituitary adenoma association. Family history and a search for associated syndromic manifestations can help to draw attention to genetic causes; many of these are now tested as part of gene panels. Identifying genetic mutations allows appropriate screening of associated comorbidities as well as finding affected family members before the clinical manifestation of the disease. This review focuses on germline and somatic mutations predisposing to acromegaly and gigantism.

## 1. Introduction

Acromegaly is a rare, chronic disorder caused by excessive growth hormone (GH) production. Common clinical manifestations include changes in appearance, headache, joint pains as well as serious systemic complications such as metabolic, cardiovascular and osteoarticular comorbidities especially axial arthritis and higher risk of tumour growth (e.g., colon polyps and thyroid nodules) [1]. Cardiovascular diseases and cancer are mostly responsible for an increased mortality in untreated patients [2,3]. Due to complications, quality of life is significantly reduced [4]. In childhood and adolescence, an excessive GH secretion before complete epiphyseal closure leads to gigantism, characterised by abnormally tall stature. The prevalence of acromegaly is estimated between 28 to 137 per million people [5]. In most studies, females are slightly more (1:1.24) affected than males and the peak age of diagnosis is within the 5th decade of life [3,5]. The most common cause of acromegaly and gigantism is growth hormone (GH) secreting pituitary adenoma (PA), also called pituitary neuroendocrine tumour (PitNET) (Box 1), which represents approximately 9–13% of all PAs.

Box 1Pituitary neuroendocrine tumour (PitNET).In 2017 The International Pituitary Pathology Club suggested that the hormone-producing cells of the pituitary are a part of the neuroendocrine system and sometimes show invasive growth, therefore, proposed to use the phrase pituitary neuroendocrine tumour (PitNET) rather than pituitary adenoma, to highlight the similarity with other neuroendocrine neoplasms [6]. This suggestion has been met with some controversy [7,8,9]. It was suggested that there is a risk that aligning adenohypophyseal tumours to other neuroendocrine tumours would raise unnecessary anxiety in patients and physicians less familiar with the disease, and for the time being suggested to carry on using the term adenoma with further discussion invited on this issue [10]. As PitNET is a valid term, in a scientific publication its use can be deemed appropriate. We acknowledge that both terms have advantages and disadvantages, and will use both terms in the review.

Pituitary hyperplasia is encountered less commonly, mainly as part of genetic disorders such as Carney complex (CNC), McCune–Albright syndrome (MAS) or X-linked acrogigantism (XLAG). In rare cases (less than 1%), neuroendocrine tumours producing growth hormone releasing hormone (GHRH) or ectopic GH-secreting tumours have been described [11,12,13]. Altered growth hormone regulation resulting in GH excess can accompany neurofibromatosis type 1, associated with optic pathway gliomas (OPG) [14]. Additionally, deficiency in the immunoglobulin superfamily member 1 (IGSF1), may result in somatotroph neurosecretory hyperfunction in adults [15].

Most somatotroph PitNETs develop sporadically; however, in nearly 46–49% of gigantism, the identifiable genetic background has been reported [16,17]. Hereditary GH-secreting pituitary tumours can manifest as an isolated manifestation, called familial isolated pituitary adenoma (FIPA), due to either loss-of-function mutations in aryl hydrocarbon receptor interacting protein (AIP) or due to gain-of-function gene duplication in *GPR101*, causing XLAG. Hereditary pituitary tumours can also be part of syndromic disease accompanied by other manifestations, often tumours of other endocrine organs, such as in multiple endocrine neoplasia type 1 (MEN1), multiple endocrine neoplasia type 4 (MEN4), MAS, CNC, or phaeochromocytoma/paraganglioma (PPGL)-pituitary adenoma association [18,19,20,21,22,23,24,25] (Table 1).

## 2. Germline Mutations

### 2.1. GH Excess as an Isolated Pituitary Adenoma, FIPA

Familial isolated pituitary adenoma (FIPA) is the most common cause of familial acromegaly. The exact prevalence of FIPA is not yet known, but it appears to be more common than initially believed (to date, over 500 families have been described) [27,28,29]. FIPA is a genetically heterogenous condition, which, in its most typical form, is characterised by the occurrence of two or more cases of PAs in one family in the absence of other associated syndromic features. However, due to de novo mutations or lack of known family history, often as a result of low disease penetrance, FIPA can also be present in apparently sporadic—so called simplex—patients or in patients with mosaicism. FIPA families can be divided into two subgroups: homogeneous FIPA, where all affected family members present the same subtype of pituitary tumour, or heterogeneous FIPA, where a combination of different PA subtypes occur in the same kindreds. Most *AIP* mutation-positive families have GH excess, in a homogenous form in 51% and within the heterogenous form with prolactinomas in 24% and usually small NFPAs in 21% [28]. There is only one *AIP*- positive homogenous prolactinoma family described to date [30]. Depending on genetic background, FIPA can be divided into three subgroups: (i) *AIP* mutation-positive patients; (ii) families with duplication of *GPR101* (all XLAG cases have GH excess, with the vast majority combined with prolactin excess); (iii) families with no identifiable genetic cause. The latter represents the largest subgroup of FIPA [18,31,32]. In our cohort of 318 *AIP* (and *GPR101*) mutation-negative FIPA families, 21% have homogenous acromegaly (representing 46% of the 147 homogeneous *AIP* negative families) and 32% has heterogenous FIPA with at least member with acromegaly (59% of the 171 heterogenous FIPA kindreds). Within the heterogenous group (171 kindreds), 29% has acromegaly and prolactinoma, 26% has acromegaly and non-functioning tumour and 4% has acromegaly and Cushing’s disease. [28]. It is impossible to establish the exact penetrance of *AIP*-negative FIPA. Comparing *AIP*-positive and *AIP*-negative FIPA families, the number of affected subjects is significantly lower in *AIP* negative kindreds, suggesting that there is lower penetrance.

#### 2.1.1. Aryl Hydrocarbon Receptor-Interacting Protein (AIP)

**Overview:** *AIP* mutations can be identified in up to 40% of familial acromegaly and gigantism [18,33,34]. These germline mutations occur in 10% of FIPA in our large FIPA cohort, although previous smaller studies suggested up to 20% [28,31]. However, *AIP* mutations are also identified in apparently sporadic (simplex) cases of PitNETs, mostly among young-onset patients. This phenomenon is observed due to low penetrance (12.5–30%) of the disease in *AIP* mutation carriers [18,31,35,36], rather than de novo mutations [18]. Current data suggest that the nature of *AIP* mutation (truncating or non-truncating) does not have any effect on its penetrance [18]. In acromegaly/gigantism patients associated with *AIP* mutations, a higher GH level has been observed, with no difference in insulin growth factor 1 (IGF-1) level [28,34] and prolactin co-secretion [18].

*AIP*-positive GH-secreting PAs manifest earlier in comparison to *AIP-*negative familial [36] and sporadic cases [34]. Typically, the onset of symptoms of the *AIP*-positive pituitary tumours in patients is before the age of 30 years old (65% of patients develop symptoms <18 years) [28]. Pituitary tumours are observed, on average, 8 years earlier in this group than in the *AIP* negative group [28]. At the time of diagnosis, patients often present larger tumours with extrasellar extension, more aggressive behaviour and a higher rate of pituitary apoplexy, especially among children [18,34,36,37,38]. Clinically, *AIP*-positive patients with GH excess are taller than *AIP*-negative counterparts from FIPA families [28]. Most studies identified more males than females (60% vs. 40%) [28,34,39], but the gender imbalance might result from ascertainment bias for genetic testing in males due to the higher prevalence of gigantism related to a longer puberty process among boys.

Familial cascade screening can lead to early diagnosis in patients with little or no obvious symptoms [28]. Prospectively diagnosed *AIP*-positive PAs are mostly less invasive and present better clinical outcomes [28]. At the time of diagnosis, they are often microadenomas, with lower rates of suprasellar extension and cavernous sinus invasion that correlate with the reduced rate of active diseases. These results highlight the clinical value of genetic testing for *AIP* mutations among acromegalic patients and their family members. On the other hand, with the identification of small, non-functioning lesions, probably representing incidentalomas, screening can lead to increased anxiety and health care spending. Follow-up of these families revealed that *AIP* carriers can present on a clinical spectrum from young-onset severe cases or slowly developing cases to patients with non-functioning stable small pituitary lesions not dissimilar to incidentalomas in the general population. Further observational studies are required to assess the cost–benefit ratio of the follow-up options.

To date, no somatic *AIP* mutation has been described [40]. However, in some studies, sporadic somatotroph tumours show low expression of AIP [41,42].

**Genetics:** The *AIP* gene was first described in 1996 as a negative regulator of the hepatitis B virus X protein [43]. The association between *AIP* mutations and pituitary tumours was found ten years later in 2006 in North-Finnish and Italian kindreds [35]. Subsequently, mutation in the *AIP* gene was noted as the most common genetic cause of FIPA, including familial acromegaly and gigantism cases [38]. Several sets of founder mutations have been identified, such as the original cohort of Finnish patients (Q14* mutation) [35], but also Italian (R304*) [44], English (a small duplication mutation) [45] and Northern Irish (R304* independent from the Italian) cohorts, the latter providing a genetic background to the historical and folklore Irish giant legends [46,47].

The *AIP* gene is a well-conserved co-chaperone protein. It has numerous partners [48], including several heat shock proteins such as heat shock protein 90 (HSP90) and heat shock cognate 70 (HSC70). Another binding partner of AIP protein is phosphodiesterase subtype 4A5 (PDE4A5), an enzyme associated with the degradation of cyclic adenosine monophosphatase (cAMP). *AIP* mutation results in elevated concentrations of cAMP [36,37]. Disrupting the cAMP pathway is an important factor contributing to pituitary tumourigenesis observed in CNC, MAS and XLAG. Dysfunction of AIP protein leads to reduced Galphai-2 and Galphai-3 protein expression, which is responsible for inhibition of cAMP synthesis. AIP loss is also associated with the loss of cell cycle regulator ZAC1, which plays a role in somatostatin-related pathways [49,50]. Therefore, decreased phosphodiesterase function, G protein dysfunction and lack of ZAC1 all could play a role in the characteristic resistance to somatostatin analogues (SSA) [51].

While truncating mutations in *AIP* are obviously disease-causing, it is a challenge to predict pathogenicity of missense variants. Various in silico as well as in vitro or in vivo experimental approaches have been tried to support clinical genetic decisions regarding these variants [36,37,52,53,54].

**Diagnosis:** Genetic testing includes sequencing in tumour suppressor genes. If negative, multiple ligation probe amplification (MLPA) should be performed. In *AIP*-positive FIPA, genetic screening of kindreds should be performed as soon as the family agrees, but not later than 4 years of age, based on the age of diagnosis of the youngest described patient [55]. Up to 10 years of age, physical examination and regular body height measurement should be performed [56]. Pituitary MRI is suggested to be performed first at the age of 10 years, and then every 5 years until the age of 30. As most cases develop symptoms before the age of 30 years, in asymptomatic *AIP* mutation carriers, follow-up is suggested to be performed until this age. Then, if no pituitary pathology has been detected, the follow-up can be relaxed or discontinued. The high frequency of pituitary incidentalomas in the general population also should be considered in *AIP*-positive patients with normal biochemical status.

**Therapy:** *AIP*-positive, GH-secreting PitNETs present more often with sparsely granulated variant, aggressive behaviour and a poor response to somatostatin analogues [41,57]. Patients with *AIP* mutation require more often multimodal approaches including radiotherapy and reoperation [28]. *AIP*-mutated somatotropinomas have been reported to have lower somatostatin receptor type 2 (SSTR2) expression [58] and therefore lower response to first generations of somatostatin analogues [59]. However, in comparison to densely granulated variants, better clinical and biochemical responses to pasireotide have been observed [60]. More recently, miRNAs have been found as predictors of tumour invasiveness and therapy outcomes. There is evidence that *AIP*-mutated PAs present different expressions of miRNAs versus non-mutated PAs [61,62]. Upregulation of miR-34a in *AIP-*positive PAs is associated with impaired treatment response to octreotide [62].

The characteristic of PAs in magnetic resonance imaging (MRI) may also predict treatment response, as sparsely granulated variants correspond with T2 hyperintensity [63,64].

#### 2.1.2. X-Linked Acrogigantism (XLAG)

**Overview**: XLAG is a recently described disease caused by either germline or somatic duplications of the *GRP101* gene [32]. The prevalence of XLAG varies between 7.8–10% of gigantism patients with female predominance (2/3 of the cases) [16,65,66,67,68,69,70,71,72]. To date, less than 40 cases of XLAG have been described. XLAG is the second most common genetic cause of childhood onset of acromegaly after *AIP*-mutated somatotropinomas [16,17,68]. The phenotype of XLAG includes non-syndromic gigantism with the presentation of the disease before the age of 5 years old. Generally, children are born with normal body length and weight, but during the first 2 years of life, accelerated growth velocity is the most prominent feature. Other observed manifestations are acral enlargement, coarse facial features, headaches and sweating [17,32]. A possibly distinguishing feature between XLAG and other pituitary gigantism cases is increased appetite, observed in one-third of patients. Fasting hyperinsulinemia has been noted in 1/3 of cases, and 20% of patients had acanthosis nigricans. Less frequently, sleep apnoea, extensive perspiration or abdominal distension have been observed [16,65,66,67,68,69,70,71,72].

Pituitary pathology varies between XLAG patients from large tumours to pituitary hyperplasia. Most patients develop mixed somatotroph/lactotroph macroadenoma with a lower tendency to local invasion and pituitary apoplexy than patients with *AIP*-mutated tumours [68]. Concomitant hyperprolactinaemia has been noted in over 85% of XLAG patients [68]. The literature describes pituitary hyperplasia in around 25% of cases. The potential cause of pituitary hyperplasia may result from the early onset of prenatal exposure to increased GHRH levels. In plasma, circulating GHRH levels can be normal or slightly elevated and in some patients, a paradoxical response in the thyrotropin-releasing hormone test has been noted [17]. Histopathologically, pituitary tissue is characterised by sinusoidal and lobular architecture and contains densely or sparsely granulated somatotrophs with microcalcifications and follicle-like structures [66,68]. In most cases a low Ki-67 index has been observed.

**Genetics:** *GRP101* gene is located in the X26.3 region. The exact mechanism of GPR101 overexpression in pituitary tumourigenesis is not fully understood. GPR101 can lead to activation of an orphan G protein-coupled receptor and increased cAMP levels, which is a key factor involved in GH secretion and cell proliferation in response to GHRH [32,68]. To date, all females have been shown to have de novo germline *GPR101* duplication [32,65,68], while mosaic mutations have been described in males except for a few familial cases with mother-to-son inheritance [65,68,70,73]. The phenotype of patients with somatic and germline *GPR101* duplication remains the same [65,68,70,73].

**Diagnosis:** Genetic testing should be performed using array comparative genomic hybridisation (aCGH) array, but in negative cases with a suggestive phenotype, alternative methods such as copy number variation digital droplet polymerase chain reaction (PCR) for *GPR101* to detect smaller duplications [68] or high-density aCGH should be considered. On suspicion of a mosaic XLAG mutation, analysis of affected tissue should be performed. Preimplantation diagnosis or prenatal screening should be considered in affected mothers, as full penetrance in familial XLAG has been observed [65,73,74].

**Therapy:** The treatment of XLAG patients remains challenging and often requires a multimodal approach [65,68]. Neurosurgery is the first line treatment among patients with pituitary tumours but often, further control of the disease requires additional medical therapy or radiotherapy. In cases of pituitary hyperplasia, total hypophysectomy could be an effective surgical treatment with the obvious disadvantage of complete hypopituitarism [75]. In patients not controlled by surgery, pegvisomant alone or combined with somatostatin analogues or dopamine agonists is an effective treatment and successfully controls linear growth [65,72,76].

### 2.2. Acromegaly as a Part of Systemic Disorder

#### 2.2.1. Mutliple Endocrine Neoplasia Type 1 and Type 4 (MEN1 and MEN4)

**Acromegaly in MEN1**: One fourth of PitNETs related to MEN1 are GH-secreting tumours [26,77,78,79]. Among all MEN1 patients, acromegaly occurs in about 10% of cases. Conversely, *MEN1* mutations have been described in 1.2% of sporadic acromegaly patients younger than 30 years [80] The prevalence of patients with acromegaly and MEN1 phenotype (defined as occurrence of at least one other MEN1-associated tumour) has been noted in 6.6% of 414 patients with acromegaly, but the prevalence of *MEN1* mutations in this group is much lower [81]. The probability of positive genetic results rises with the occurrence of three types of endocrine tumours [81]. 

The prevalence of primary hyperparathyroidism among patients with acromegaly is higher than in the general population (6.1% vs. 0.86%) [81,82]. In several studies, patients with acromegaly presented increased calcitriol levels and fibroblast growth factor 23. However, the exact mechanism of this relationship between GH excess and hyperparathyroidism has not been clarified yet [83]. 

The age at the diagnosis of acromegaly in the course of MEN1 is around 40 years [84]. GH-secreting PAs in MEN1 patients are often macroadenomas with local invasion, plurihormonal profile and poor response to medical treatment but still better clinical outcomes in comparison to *AIP*-mutated PAs [85]. Pituitary hyperplasia alone or coexisting with a pituitary tumour is more common in patients with MEN1/MEN4 compared to *MEN1*-negative tumours [86]. In some MEN1 patients, poorly differentiated PIT1-lineage tumours, previously known as “silent subtype 3 adenoma”, have been observed [87], with a variable combination of GH, prolactin, α-subunit and thyroid-stimulating hormone. In patients with acromegaly and MEN1 syndrome, GHRH-secreting pancreas tumours should be considered [88]. Ectopic GHRH and GH production due to lung neuroendocrine tumour related to MEN1 mutation has been found only in one patient [89]. Gigantism associated with *MEN1* mutation occurs in approximately 1% of cases [16], this could be due to a pituitary tumour or, rarely, due to a GHRH-secreting pancreas tumour [90]. Possible coexistence of acromegaly due to pancreatic GHRH excess and prolactin-secreting or non-functioning pituitary tumour remains a diagnostic challenge in MEN1 patients.

**Overview**: PitNETs occur in 30–40% of patients with MEN1 syndrome, in addition to hyperparathyroidism (95–100% of cases) and pancreatic neuroendocrine tumours (60% of cases). Other common but non-endocrinological manifestations (up to 85% of cases) include cutaneous skin lesions (angiofibromas, collagenomas, café-au-lait macules) [84,91]. PAs in general could be the first manifestation in about 20% of MEN1 cases, and many of these are in childhood or adolescence. Some authors suggest screening *MEN1* gene in this age group [80,92]. *MEN1* mutated PAs manifest predominantly in the 4th decade of life, but various ages of onset have been noted (from 5 years to 90 years). The most common clinically presenting pituitary tumour type is prolactinoma (60% of cases), followed by non-functioning pituitary adenoma (NFPA) and somatotropinoma [26,77,78,79]. However, recent studies have shown increased numbers of NFPA among asymptomatic MEN1 patients as a result of family cascade screening [93]. The phenotype of prospective diagnosed PAs is similar to sporadic cases.

**Genetics:** Inactivating mutation of the *MEN1* gene, located on chromosome 11q13, was first reported in 1997 [94], but the phenotype of MEN1 syndrome was first noted in a patient with acromegaly and enlarged parathyroid glands by Erdheim in 1903. The *MEN1* gene contains of 10 exons and encodes a 610 amino acids protein, menin [95]. To date, over 1800 pathogenic gene variants have been described [96,97]. More recently, *MEN1* mosaic mutations have also been reported [98,99]. Most pathogenic germline *MEN1* variants are frameshift mutations (42%), followed by nonsense mutations (14%), missense mutations, splice site mutations and large deletions [96,97]. Inactivating mutations of *MEN1* lead to premature menin truncation and its impaired activity. The menin function has been proven in cell proliferation, cell signalling, transcriptional regulation and genome stability Nevertheless, its role in tumourigenesis has not been fully understood [84,97,100]. Menin, as a member of the histone methyltransferase complex, regulates the expression of the cyclin-dependent, kinase-inhibiting genes (*CDK*), *CDKN1B* (encoding p27) and *CDKN2C* (encoding p18) and possibly other *CDK* inhibitors [101,102]. The association between the *MEN1* gene and *CDK1B* may explain a similar phenotype of MEN1 and MEN4 syndrome.

**Diagnosis:** Diagnosis of MEN1 could be (i) clinically established if a patient develops two or more MEN1 associated tumours (pituitary and parathyroid adenoma, pancreatic neuroendocrine tumour); (ii) by the presence of one characteristic MEN1 tumour and one first-degree relative with confirmed *MEN1* mutation or (iii) due to family cascade genetic screening in asymptomatic carriers [77]. No direct genotype–phenotype correlation of MEN1 has been confirmed. The importance of genetic testing has been established for an early diagnosis and the identification of asymptomatic carriers. Genetic screening tests should be aimed to search for sequence variation or large deletions. It is suggested to start genetic screening at the age of 5 years [91,103]. *MEN1* mutation carriers should undergo periodic clinical screening. At the time of diagnosis, baseline biochemical, pituitary and abdominal imaging should be performed and then repeated at 1–3 year intervals. Yearly clinical and biochemical (serum calcium, gastrointestinal hormones, prolactin and IGF-1) assessment is advised. Abdominal and pituitary MRI in asymptomatic mutation carriers should be performed first at the age of 10 years [84].

**Therapy:** First line treatment for MEN1-associated acromegaly is the same as in current acromegaly guidelines [77]; however, patients require neurosurgery more often and multimodal approaches, especially paediatric cases. Recent studies of *MEN1* animal models brought up important knowledge of tissue-specific tumorigenesis mechanisms of menin and enabled testing new treatment strategies [104,105]. It has been suggested that *MEN1* gene replacement, by the use of adenoviral vectors, would decrease pituitary tumour proliferation. Another option is the potential use of a monoclonal antibody to the vascular endothelial growth factor (VEGF-A), which inhibits angiogenic pathways. In a study of MEN1 mouse models with prolactinoma, the implementation of VEGF-A resulted in lowering of the prolactin concentration in treated animals but not controls. Blockade of angiogenesis may be considered as a nonsurgical treatment option for benign, endocrine tumours associated with MEN1 syndrome [106].

MEN4: About 10–20% of patients presenting a MEN1-like phenotype have no identifiable *MEN1* mutations. Further genetic investigations have revealed a small number of patients harbouring loss of function mutation in the *CDKN1B* gene (up to 3% of cases with negative *MEN1* results) [107,108,109]. The syndrome of association between MEN1 phenotype and *CDKN1B* mutations has been termed MEN4. The *CDKN1B* gene, located on chromosome 12q13, encodes p27 and regulates the cell cycle. To date, less than 50 cases with *CDKN1B* mutations (the majority of patients presenting with hyperparathyroidism) have been noted, one-third of those conjoined with pituitary tumours [110,111,112,113]. Due to the rarity of the disease, penetrance and genotype–phenotype correlation cannot be assessed yet. Somatotroph and corticotroph PAs have been found to be the most common pituitary tumour among symptomatic MEN4 patients. In *MEN4* mutation carriers found by family screening, NFPA has been described as the most common [110,111,112]. In a large cohort of 190 patients with Cushing’s disease, 2.6% had *CDKN1B* variants [113].

In rare MEN1 and MEN4-like patients with negative genetic results, other *CDKI*s pathological variants (p15 [*CDKN2B*, 1%], p18 [*CDKN2C*, 0.5%], p21 [*CDKN1A*, 0.5%]) should be considered [114], as well as *CDC73* gene mutation [81,115,116] (responsible for hyperparathyroidism-jaw tumour syndrome) and *CaSR* mutation [115,116] (causing familial hypocalciuric hypercalcemia).

#### 2.2.2. McCune–Albright Syndrome (MAS)

**Acromegaly in MAS:** GH-secreting pituitary tumours or pituitary hyperplasia are the most common pituitary disease associated with McCune–Albright Syndrome. Acromegaly is present in up to one-third of cases [117]. The mean age of onset of GH excess in MAS is observed in the 2nd decade of life, and the incidence is significantly more frequent in males (75%). Concomitant hyperprolactinaemia occurs in 71–92% of acromegaly cases. Gigantism related to MAS has been described in 5% of patients with GH excess with childhood-onset [16]; however, body height in this syndrome is also dependent on precocious puberty, which increases bone age; therefore, this peculiar disease can be the cause of both gigantism and abnormal short stature.

**Overview****:** MAS is classically characterised by a triad of fibrous dysplasia, precocious puberty and café-au-lait skin lesions [118]. The prevalence of MAS is estimated between 1/100,000 and 1/1,000,000 [118]. The variety of endocrinological manifestations in addition to GH excess includes hypercortisolaemia (due to nodular adrenal hyperplasia), or thyrotoxicosis. The suspicion of MAS should be considered when acromegaly/gigantism is associated with other syndromic features of this disease. Probably, the oldest known case of MAS is the Tegernsee giant, who died in 1876. He presented juvenile gigantism (body height 230 cm) with concomitant fibrous dysplasia [119].

**Genetics**: MAS is caused by mosaicism for mutations in *GNAS* gene, located at chromosome 20q13.3. The phenotype of MAS is dependent on the cell type and the number of affected tissues. The *GNAS* gene encodes the stimulatory α subunit of guanine nucleotide-binding protein [120,121]. A gain-of-function mutation in the *GNAS* gene, affecting codons Arg201 and Gln227, results in a constitutively activated cAMP pathway and leads to persistent GH hypersecretion and cell proliferation. [122,123,124]. The consequences in the pituitary include overproduction of GH and sometimes also prolactin and hyperplasia or tumour.

**Diagnosis:** The diagnosis can be made clinically by a complete physical and biochemical evaluation of patients. When the clinical, radiological and histopathological analysis is unclear, genetic testing should be performed. Molecular diagnosis could include Sanger sequencing of samples of affected tissue, and while this technique had a lower sensitivity from peripheral blood lymphocytes, more recently, digital droplet PCR from whole blood or from circulating cell free DNA showed 80% sensitivity [125].

**Therapy:** First-line treatment of acromegaly/gigantism related to MAS with pituitary hyperplasia is somatostatin analogues. In resistant cases, pegvisomant alone or in combination with octreotide or lanreotide is recommended. Pasireotide has also been used in the treatment of GH excess in MAS [126]. If concomitant hyperprolactinaemia occurs, a dopamine agonist should be added as well. In patients not responding to pharmacological therapy, pituitary surgery should be considered. Neurosurgery is challenging due to concomitant skull base fibrous dysplasia, as high vascularity of these bony lesions gives a high risk of haemorrhage. If operated, total hypophysectomy is suggested, as in most of the cases, the whole gland is involved. Radiotherapy of the pituitary gland should be carefully considered for severe disease if previously therapy options have failed. Radiation of associated bone lesions may lead to malignant transformation of sarcoma [117].

#### 2.2.3. Carney Complex (CNC)

**Acromegaly in CNC**: In up to 75% of patients with CNC, the asymptomatic elevation of GH, IGF-1 and/or prolactin or abnormal response to the thyrotropin-releasing hormone is observed. Clinically evident acromegaly due to pituitary tumour occurs in 10–12% of cases, with slight female predominance [127,128]. Acromegaly was the first manifestation of CNC in four patients reported in the literature; however, at the time of diagnosis, a majority of patients had several other CNC symptoms [128]. Pituitary manifestation includes pituitary tumour, pituitary hyperplasia or a combination of both [129]. Acromegaly manifests in the 3rd decade of life [130], but gigantism related to CNC has also been noted. GH-secreting PitNETs are often multifocal, surrounded by somatomammotrophic hyperplastic tissue.

**Overview****:** CNC is a rare genetic disorder with multiple endocrine and non-endocrine symptoms, with an autosomal dominant inheritance and high penetrance for some manifestations (>95% by the age of 50). It was originally described by Professor Carney as a “complex of myxomas, spotty pigmentation, and endocrine overactivity” in 1985 [131]. However, the first (now molecularly confirmed) CNC patient was reported by Professor Harvey Cushing in 1913. The patient presented acromegaly due to pituitary tumour, skin pigmentation and adrenal pathology [132]. The analysis of archive tissue revealed a *PRKAR1A* mutation. The most common endocrinological manifestation is adrenocorticotropic hormone (ACTH)-independent Cushing’s syndrome due to primary pigmented nodular adrenal disease (PPNAD). The prevalence of Carney complex is unknown. To date, over 700 cases have been described [133].

**Genetics:** The majority of CNC cases are familial. Most are caused by a germline-inactivating mutation mainly in the *PRKAR1A* gene (CNC1), located on the 17q22-24 locus [134,135], but recently other protein kinase A regulatory subunit 1α (PKA) mutations, including *PRKACB*, have also been described [136,137,138]. In approximately 30% of cases, CNC occurs as a consequence of a de novo mutation. The study with the largest number of patients with Carney complex has found a genotype–phenotype correlation [127]. Patients with large deletions of *PRKAR1A* develop the diseases earlier with a more severe phenotype, including metastatic psammomatous melanotic schwannoma [139].

*PRKAR1A* is composed of two catalytic and 2 regulatory subunits and is implicated in transcriptional regulation, cell proliferation and apoptosis. Inactivating mutations of *PRKAR1A* lead to uncontrolled activation of cAMP-dependent kinase activity in affected tissues [135]. Carney complex 2 locus (CNC2), located on chromosome 2p16, accounts for 20% of cases; however, the responsible gene at this locus has not been found yet [140,141]. The incidence of acromegaly was similar in CNC1 and CNC2 groups.

**Diagnosis:** Diagnosis of Carney complex in a patient may be established clinically if two or more major criteria are present (characteristic skin lesions, cutaneous and heart myxomas, PPNAD, acromegaly, large-cell calcifying Sertoli cell tumour or characteristic calcification of testis, thyroid carcinoma or multiple hypoechoic nodules, breast ductal adenoma psammomatous melanotic schwannomas, blue nevus, osteochondromyxoma) [19]. Another way to confirm CNC diagnosis is the occurrence of one major criterion and an affected first-degree relative or a known inactivating *PRKAR1A* mutation. Genetic testing may be offered for patients with two major diagnostic criteria or for relatives of patients with Carney complex. Molecular techniques include Sanger sequencing. In negative cases, copy number variant analysis by CGH or deletion testing should be performed.

**Therapy**: To date, there is no specific treatment approach for acromegaly in CNC cases, and guideline on the management is the same as in sporadic cases. In a great majority of patients, surgery alone or combined with SSA has been used. If multiple pituitary tumours are present, partial or complete hypophysectomy should be performed. Some authors suggest that due to overactivation of cAMP signalling in CNC patients, the use of SSA theoretically would be beneficial. However, resistance to SSA treatment has been observed [128].

#### 2.2.4. Phaeochromocytoma/Paraganglioma (PPGL) and Pituitary Adenoma Association (3Pa)—SDHx/MAX Mutations

**Overview:** The coexistence of pituitary adenoma with PPGL was first described in 1952 in a patient with acromegaly and phaeochromocytoma [142]. Genetic predisposition of this rare condition has been relatively recently found in 2009 in a familial case of prolactinoma with paraganglioma and *SDHB* mutation [143]. Subsequently, in 2012, a patient with aggressive GH-secreting PitNETs with bilateral phaeochromocytomas and a pathogenic variant of *SDHD* mutation was described [144]. To date, <100 cases of PPGL and pituitary adenoma association, also known as “three P association” (3Pa), have been described worldwide, which represent a genetically heterogenous group [22,145,146].

**Acromegaly with *SDHx* mutation**: The most common genetic cause of 3Pa is a germline loss of function mutation of the succinate dehydrogenase (*SDH)x* gene. GH-secreting PAs associated with *SDHx* mutations tend to be aggressive macroadenomas. To date, 4 GH-PitNETs with an *SDHx* mutation have been described (*SDHD* and *SDHB* mutations). Gigantism related to *SDHx* mutation has not been reported yet. Three patients harboured macroadenomas (data about tumour size was not available in the 4th case), and they were treated with SSA alone or combined with surgery. The age at the diagnosis of acromegaly varies from 37 to 84 years old [147]. A unique histopathological feature of pituitary adenoma with *SDHx* mutations is intracytoplasmic vacuoles, which can correspond to the presence of autophagic bodies [148]. More recently, *SDHx* mutations have been observed in patients with an isolated pituitary tumour and without personal or familial history of PPGL, but none of them had somatotropinoma (3 prolactinomas out of 263 patients with PAs) [146].

**Genetics:** There are several genes encoding the SDH protein complex (*SDHA*, -*B*, -*C*, -*D* or *SDHA2F*). This multimeric enzyme plays a crucial role in tricarboxylic acid or the Krebs cycle and respiratory chain. Germline mutation of *SDH* results in the accumulation of oncometabolites that inhibit degradation of hypoxia transcription factor (HIFα) [149]. The penetrance of pituitary tumours in *SDHx* mutation-positive patients is very low (1% of cases).

**Acromegaly with *MAX* mutation:** The association of PPGL and PA has been described in a few cases due to MYC-associated factor X mutations (*MAX*). They were somatotroph macroadenomas, including a childhood-onset case, and prolactinomas [150,151]. Patients required multimodal approach, as surgery alone was not sufficient. The combination of SSA with cabergoline and pegvisomant, as well as radiotherapy, was used [150]. Patients with germline *MAX* mutation may also develop other systemic manifestations like renal oncocytoma or lung cancer [152].

**Genetics:** The *MAX* gene is located on chromosome 14q23.3. MAX interacts with other parts of the MAX-MLX network, which is responsible for the integration of cellular signals and modulates the expression of another gene [153]. Germline *MAX* mutations are associated with tumourigenesis involving neuroendocrine cells, renal tumours or small cell lung cancer [152,154]. Point mutations and small exonic and intronic deletions [150,151] of *MAX* have been linked to PAs.

The association of pituitary tumours and PPGL may also appear due to *MEN1* mutation (to date, one mixed GH/PRL macroadenoma out of four PitNETs [148]). NF1 can be associated with PPGL and GH excess (see above). Other germline mutations, including *RET* and *TMEM127*, have also been described with pituitary adenomas, but more data are needed to asses if they are indeed involved in pituitary tumourigenesis or if these cases are coincidences. [148,155,156]. A rare clinical situation of acromegaly due to ectopic GHRH production by PPGL (usually phaeochromocytoma) has also been described [157,158], including a *MAX* mutation positive case [159].

### 2.3. Other Syndromic Disease Associated with Germline Mutations and GH Excess without Visible Pituitary Tumour/Pituitary Hyperplasia

#### 2.3.1. Neurofibromatosis Type 1 (NF1)

**Acromegaly in NF1**: Clinical features of acromegaly and gigantism with GH excess have been observed in 10% of children with NF1 and symptomatic OPG without visible pituitary tumour. In these patients, the OPG involved the chiasm and reached the optic radiations and temporal regions. The specific mechanism of *NF1* mutations leading to GH excess has not been identified yet [14,160]. The causes of GH excess may result from the loss of hypothalamic somatostatinergic inhibition, increase in GHRH stimulation, and the role of activation of hypothalamic GPR101 is also possible. Interestingly, the GH excess often improves later in life. Somatostatin analogues and pegvisomant have been reported as an effective treatment in patients with NF1 and GH excess [14,161,162]. Normalisation of GH after SSA treatment has been observed [160,163]. NF1 and *bona fide* pituitary adenoma is extremely rare and possibly a coincidence, only reported in two patients [163,164], including a 68-year-old female with somatotropinoma, hyperparathyroidism and follicular thyroid carcinoma. Genetic testing confirmed *NF1* mutation and excluded *MEN1* mutation. The PA tissue showed no loss of the wild type allele of the *NF1* gene, but harboured a somatic *GNAS* p.R201C mutation, not supporting NF1 being causative in pituitary adenoma development. In the other published case with *NF1* mutation and somatotroph PA, somatic changes were not assessed [164].

**Overview****:** NF1 is one of the most common genetic disorders, with the prevalence estimated at around 1:2500–1:3500 live births. This autosomal-dominant syndrome is caused by an inactivating mutation of the *NF1* gene, located on chromosome 17q11.2. It encodes neurofibromin, a protein involved in cell growth and proliferation, by inhibiting RAS activity and regulation of cAMP levels [165]. The most common characteristic features are cutaneous neurofibromas, cafe-au-lait skin lesions, intertriginous freckling, Lisch nodules and brain tumours, including the most common optic pathway glioma [166]. The diagnosis can be made clinically if the patient presents two or more signs of the condition.

#### 2.3.2. Deficiency of the Immunoglobulin Superfamily Member 1 (IGSF1)

Recent data revealed that IGSF1 deficiency results in somatotroph neurosecretory hyperfunction [15]. The *IGSF1* gene, located on the X chromosome, is highly expressed in the hypothalamus and pituitary. Its loss-of-function mutations cause central hypothyroidism, hypoprolactinaemia and macroorchidism. Additionally, 52.4% of adult patients with germline *IGSF1* mutation present acromegalic facial features as well as organ changes due to GH excess. Tall stature does not occur. The average age of onset of GH hypersecretion has not been defined yet. Biochemically, the IGF-1 level rises usually above the mean (above 1 SDS). A germline *IGSF1* variant has been identified in three family members with XLAG-related gigantism, but the reported variant has up to 0.01 minor allele frequency and has been reported as benign by ClinVar, and is therefore unlikely to be related to the phenotype [167]. More research is required to define the exact role of IGSF1 in the regulation of the somatotroph axis.

#### 2.3.3. Tuberous Sclerosis Complex (TSC)

Tuberous Sclerosis Complex is an autosomal-dominant genetic disorder caused by loss-of-function mutation in either the *TSC1* gene on chromosome 9q34.13 or the *TSC2* gene on chromosome 16p13.3. The phenotype of the disease includes multiple hamartomas of the brain, lungs, heart, skin and kidney. To date, four patients with the *TSC* mutation and pituitary tumour have been described, including only one somatotropinoma [168,169,170,171]. More data are needed to determine the exact role of TSC in pituitary tumourigenesis.

## 3. Somatic Variants in GH-Secreting PitNETs

### 3.1. GNAS

The first identified and the most common somatic mutation found in isolated GH-producing PitNETs is in the *GNAS* gene at the 201 and 227 codons [172,173], with a frequency of 30–40% of sporadic acromegaly (range 10–50% depending on the ethnicity of patients) [174]. *GNAS*-mutated somatotroph tumours are smaller and less likely to be invasive but contribute to higher GH and IGF-1 levels. In comparison to *AIP* and *GPR101* mutated PAs, *GNAS*-positive tumours arise in older patients and show a better response to first-generation somatostatin analogues treatment [175]. Furthermore, in *GNAS*-mutated PAs, higher expression of dopamine receptor 2 has been observed. *GNAS* status could have a potential value in predicting better treatment response to dopamine agonists [172]. However, in a study of genome-wide sequencing, *GNAS* mutation has been found in 5 out of 8 plurihormonal PAs secreting GH and prolactin and in 9 out of 23 pure GH somatotropinomas. Alterations of DNA methylation have also been linked with *GNAS* mutation [172]. In a recent study, PIT1 lineage tumours showed global hypomethylation, chromosome alterations and transposable element overexpression [172]. A negative correlation between DNA hypomethylation and chromosome instability has been observed. However, in *GNAS* mutated tumours, DNA hypomethylation and limited chromosomal alterations have been noted. In *GNAS* wild-type GH-PitNETs, unexpectedly, gonadotroph marker (SF1) expression has been described [172].

To date, no association has been observed between *GNAS* mutation and granulation patterns in histopathology results [176,177]. *GNAS* is an imprinted gene with maternal allele expression in normal pituitary tissue. In *GNAS* positive patients, mutations are almost always located on the maternal allele due to paternal imprinting [178,179]. Somatic mosaicism of the *GNAS* gene results in the previously described McCune–Albright syndrome.

### 3.2. Glucose-Dependent Insulinotropic Polypeptide Receptor (GIPR)

In around 30% of GH-secreting PAs with negative *GNAS* mutation, GIPR is expressed at a significantly higher level than in the normal pituitary gland. Clinically, in these patients, paradoxical increase in GH after oral glucose load test has been observed [180]. In a study analysing 496 patients with acromegaly, a paradoxical response was associated with older age at the diagnosis, smaller and less invasive PAs and better treatment outcomes [181]. The potential mechanism could be explained by increased gastric inhibitory peptide (GIP) stimulation and therefore higher GH levels in the glucose load test. The GIP/GIPR axis stimulates GH secretion by mimicking the cellular pathways triggered by GHRH stimulation [180,182,183]. A paradoxical rise of GH to oral glucose in acromegaly may help to predict clinical characteristics of PitNET that may influence therapy approaches [184]. However, other biological factors, like BMI, oral oestrogen intake, diabetes mellitus or hyperthyroidism should be taken into account when interpreting GH values after glucose suppression [185].

### 3.3. Other Genes

Studies using whole-genome sequencing and whole-exome sequencing did not find any consistently recurrent somatic mutations. However, several somatic variants associated with the cAMP pathway, calcium signalling and ATP signalling have been observed (Table 2), which may suggest the important role of these pathways in the pathogenesis of GH-secreting PAs [186,187].

In patients with *DICER1* mutations and pituitary blastoma, immunohistochemistry for GH was positive in 10 out of 14 studied tumours. Biochemically, serum GH and IGF-1 levels were not elevated. Clinically, patients did not present increased growth velocity or tall stature [188].

In a significant proportion of cases, the genetic background of pituitary tumourigenesis is still not fully understood. Further studies found a relationship between epigenetic modifications and pituitary tumourigenesis. Epigenetic alterations may occur at chromatin levels such as DNA methylation and histone modification, or via non-coding RNAs, microRNAs or proteomics [189]. Also, the tumour microenvironment, including a variety of non-neoplastic and non-cellular elements, may modulate pituitary tumourigenesis [190,191].

## 4. Recommendations for Genetic Screening in Acromegaly and Gigantism

Main determinants of genetic testing in patients with acromegaly and gigantism are the age of onset of symptoms, pituitary tumour type (pituitary alone or with concomitant hyperplasia, histopathology subtype), family history or manifestations of syndromic diseases in patients or their family members (Figure 1) [192].

While today the majority of genetic testing in genetically heterogeneously determined diseases is based on gene panel testing, we can draw characteristic clinical aspects pointing to specific characteristic genes. In infant-onset GH excess (usually already manifesting before the age of 1 year, but all cases by age of 5), X-linked acrogigantism should be considered. If blood-derived *GPR101* duplication testing is negative using a CGH array, analysis of affected tissue or alternative tissue DNA and gene-specific methods (ddPCR) should follow before ruling out this diagnosis. In cases of childhood-onset of GH, excess *AIP* mutations represent the highest likelihood. While most cases are macroadenomas at clinical presentation, a few microadenomas have also been described. MEN1 syndrome-related childhood-onset GH-secreting adenoma or GHRH-secreting tumours are rare but described. *MEN1* and *CDKN1B* testing is recommended in patients with personal or family history of kidney stones, neuroendocrine tumours or pituitary tumours. McCune–Albright and Carney complex are usually clinical diagnoses, but genetic testing can confirm the clinical findings. Association with paragangliomas or phaeochromocytomas should prompt panel testing for associated genes for these diseases.

Genetic testing can provide both advantages and disadvantages. We should consider the disadvantages: (i) psychological burden of increasing anxiety, guilt and depression due to carrying a genetic alteration and transmitting it to offspring, (ii) identifying variants with uncertain significance leading to uncertainty and (iii) costs. Advantages, however, usually outweigh the disadvantages: (i) to family members for early diagnosis via cascade screening [18,34,39,77], enabling better prognosis, (ii) to patients to search for other syndromic manifestations, (iii) to provide an explanation for the disease, which patients often greatly appreciate, even if no further clinical or therapeutic advantages follow and (iv) helping to understand disease mechanisms that may lead to novel future therapies. Careful discussion and individualised decision-making help to achieve balanced clinical management.

To date, there is no universally accepted genetic panel test available for acromegaly patients, but most reference laboratories offer next-generation sequencing panel testing including several genes like *AIP*, *MEN1*, *CDKN1B*, *PRKAR1A*, *SDHA*, *SDHB*, *SDHC* and *SDHD* rather than sequencing of particular genes. If multiple endocrine neoplasia is suspected, a gene panel including *AIP*, *CDC73*, *CDKN1B*, *MEN1*, *RET* genes is recommended (https://www.exeterlaboratory.com/, 19 March 2021).

## Figures and Tables

**Figure 1 jcm-10-01377-f001:**
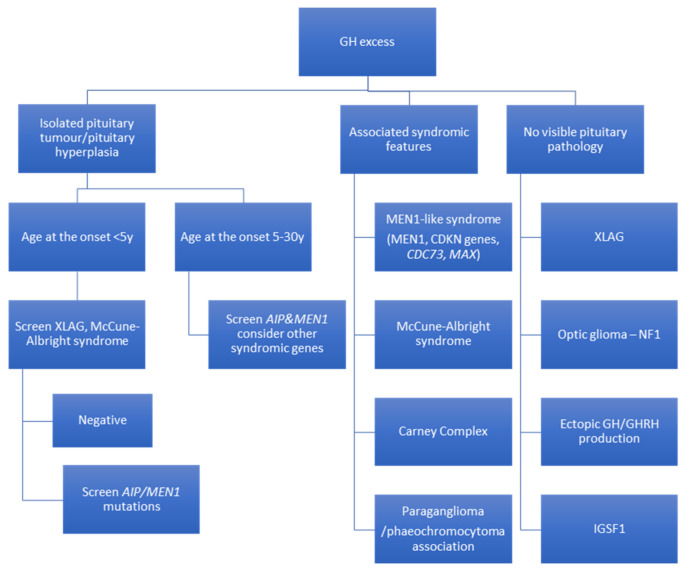
Suggested algorithm of genetic screening of GH excess. GH, growth hormone; GHRH, growth hormone releasing hormone; XLAG, X-linked acrogigantism; AIP, aryl hydrocarbon receptor-interacting protein; MEN1, multiple endocrine neoplasia type 1.

**Table 1 jcm-10-01377-t001:** Germline and somatic GNAS mutations associated with acromegaly and gigantism (adapted from Gadelha et al. [26]).

Disease	Gene Mutation/Genetic Alteration	Gene Location	Prevalence in Pituitary Tumours	Prevalence in Acromegaly (%)	Phenotype	Mean Age of Diagnosis of GH Excess	Histopathology
FIPA/AIP	*AIP*	11q13.3	3.6%	50% in homogeneous FIPA 4% in sporadic acromegaly 29% in gigantism patients	Isolated pituitary tumour	2nd decade of life (<30 years), male predominance, reduced SSTR 2 expression	More often sparsely granulated variant
FIPA/X-linked acrogigantism	*GPR101*	Xq26.3	1.6%	0–4.4% in acromegaly 10% of gigantism patients	Isolated pituitary tumour	first years of life (<5 years) female predominance, pituitary hyperplasia or tumour, males can be mosaic or familial	Often somatotroph h/lactotroph pituitary hyperplasia in 25% of cases
Multiple Endocrine Neoplasia type 1	*MEN1*	11q13.1	0.6–2.6%	1.2% in acromegaly 1% of gigantism patients	Hyperparathyroidism, pituitary tumour, pancreatic neuroendocrine tumours	4th decade of life female predominance	Multiple PAs and more often plurihormonal profile. More often pituitary hyperplasia. In some part of patients, poorly-differentiated PIT1- lineage tumours
Multiple Endocrine Neoplasia type 4	*CDKN1B*	12p13.1	rare	rare	Hyperparathyroidism, pituitary tumour, pancreatic neuroendocrine tumours	Single cases	More often pituitary hyperplasia
McCune–Albright Syndrome	Mosaic *GNAS* mutation	20q13.3	Only acromegaly/gigantism (20% of patients)	5% of gigantism patients	Classic triad: fibrosus dysplasia, cafe- au-lait macules, precocious puberty	2nd decade of life male predominance, pituitary hyperplasia, prolactin cosecretion	More often pituitary hyperplasia
Carney Complex	*PRKAR1A*	17q22-24	Only acromegaly/gigantism (12% but 75% asymptomatic elevation of GH and IGF-1	1% among gigantism patients	Acromegaly, cardiac and cutaneous myxomas, PPNAD, lentiginosis	3rd decade of life no gender predominance, hyperplasia (majority) or tumour	somatotroph h/lactotroph pituitary hyperplasia
	CNC2 locus	2p16					
Pituitary adenoma and PPGL association	*SDHx* *VHL* *MEN1* *RET*	SDHA 5p15.33 SDHB 1p36.13 SDHC 1q23.3 SDHD 11q23.1	rare	rare	Association between PPGL and pituitary tumour	Single cases	intracytoplasmic vacuoles
	*MAX*	14q23.3					
Neurofibromatosis type 1	*NF1*	17q11.2	Only acromegaly/gigantism- around 10% in patients with *NF1* and optic glioma	rare	Neurofibromas, cafe au-lait macules, freckling, Lisch nodules, optic glioma	No visible pituitary pathology	-
Deficiency of the X-link immunoglobulin superfamily member 1	*IGSF1*	Xq26.1	Only GH excess features	Not estimated	acromegalic facial features organomegaly in adulthood	No visible pituitary pathology	-
Sporadic somatotropinomas	Somatic *GNAS* mutation	20q13.3	Only acromegaly	40%	Isolated pituitary tumour	smaller size, good response to medical treatment with somatostatin analogues	no association has been observed between *GNAS* mutation and granulation pattern

FIPA—familial isolated pituitary adenoma; AIP—aryl hydrocarbon receptor protein-interacting protein; *MEN1*—multiple endocrine neoplasia type 1 gene; CDKN1B—cyclin-dependent kinase inhibitors 1b; PRKAR1A—protein kinase A regulatory subunit type I alpha; *SDHx*—genes of the succinate dehydrogenase family (A, B, C or D); PPNAD—primary pigmented nodular adrenal disease; GH—growth hormone; PPGL—phaeochromocytoma/paraganglioma.

**Table 2 jcm-10-01377-t002:** Somatic variants associated with somatotroph PitNETs and with the cyclic adenosine monophosphatase (cAMP) pathway, calcium signalling, and adenosine triphosphate (ATP) signalling [186,187].

cAMP Pathway	Calcium Signalling	ATP Signalling
*GNAS* *PRKAA2* *ADRBK2* *ATP6V0A1* *CCR10* *CHRM3* *OR51B4* *GNAQ*	*CACNA1H* *CAPN1* *DMD* *GRIN2B* *JPH2* *MAN1A1* *PCDH11X* *PROCA1* *SLIT2* *SPTA1* *TESC* *C2CD3* *RYR1* *SSR3* *WIPI1*	*SUPV3L1* *ATPAF2* *ATAD2B* *DICER1* *AOX1*

## Data Availability

No new data were created or analysed in this study. Data sharing is not applicable to this article.

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
