# Peer review of "Genetics of Acromegaly and Gigantism"

_jcm, 2021, doi:10.3390/jcm10071377_

Round 1

Reviewer 1 Report

The authors have written a very comprehensive review of the literature on the topic that presents all of the main causes and data.  There are a few points that should be addressed for clarity and caution:

Introduction.  PitNet is a controversial name and has led to a lot of heat and argument in the field in recent years.  Why is this term used here if the authors do not believe it to be useful:  "The PANOMEN Workshop recommends that the term adenoma be retained."  JES March 2021

Lines 99-103. The difference in gender at diagnosis is not that subtle; in some large studies its about 60/40.  In ref 13 the familial AIP mutation cases overall (?acromegaly also) are significantly more likely to be male, and the sporadics are about 60/40.  Ascertainment bias might have a role to play if there were truly a population of under/undiagnosed less severe female patients diagnosed later, but the numbers dont seem to be there.  I would suggest describing the percentages for males and females here at least.  Looking at the cited work on the large family from NI, were there not 3 males and 1 female identified with R304* in addition to two male probands?  Discounting the A299V alone carriers (which is not a pathogenic change as shown convincingly in Aflorei et al 2018), how does the cited study support equal male and female numbers of patients with AIP mutated acromegaly?

Lines 107-108.  The value of prospective screening has to be judged based on better outcomes for the investment of time and resources to diagnose at an earlier stage.  The authors note that prospectively identified adenomas in AIP mutation carriers are essentially indistinguishable from incidentalomas.  On the basis of epidemiology some of these are almost certainly  incidentalomas that have nothing to do with AIP status.  The issue of ascertainment bias was raised earlier as a critique of alternative conclusions derived from data.  Here one could say that prospective studies could be leading to over-diagnosis of incidentalomas as "AIP mutation associated" pituitary adenomas: this has a burden on patients and screening/followup costs.  Observational studies of these tumors might show that a reasonable proportion of them behave aggressively or at least differently to the benign course of incidentalomas, but that information is not available.  Some caution about the topic should be included.

Lines 116-122 need to be revised. The discovery of AIP in Finnish (and Italian) patients is dispatched with in a sentence and the later finding on the Irish giant founder years later doesnt directly follow logically.  The significant role of AIP mutations in gigantism was noted earlier than 2011.

Line 135 and earlier. The authors group has published a series of studies looking at the prediction of pathogenicity, so it would be very appropriate to describe those here in more detail.  They might also like to take the opportunity to describe work on miRNAs (their own included) that model the clinical features of AIP mutation related acromegaly including somatostatin analog resistance.

Line 159.  Where do the numbers (4-10%) quoted come from?  Two citations are case reports and the other a review. 

Line 168.  What is the evidence that overeating leads to increased BMI?

Line 190-191.  Please revise the references to include the works demonstrating the  the role of somatic mosaicism.  What is the role of ref 53 here and below when other correct references dealing with XLAG are not cited?

Lines 199-208 are poorly phrased.  The authors should present the totality of the data on surgery, there are many more and better examples than a case report.  Somatostatin analogs are presented twice, once as being useful in combination and then not being that useful at all.  It is now well demonstrated that they have little utility in this disease and pegvisomant is the best approach.

Lines 209 and onwards.  The authors need to decide what is their thinking on PitNETS?  Are they abandoning the term or supporting it? 

Line 440.  On MAX it is wrong to say large deletions as they are small exonic and intronic mutations.  Also reference 137 is about TMEM127, so it is not related to MAX. Larger rearrangements were demonstrated in relation to non pituitary tumors.

Reviewer 2 Report

The authors offer an in depth review of genetic causes of acromegaly/gigantism described and  recognized in published medical literature.

I am not familiar with the audience of this Journal but the topic very specialized.

Authors are experts in their field and provide extensive reference of literature for this topic.

The article is written in a logical fashion, includes updated data but perhaps could be shortened for the intended audience.

Suggestions in general :

Consider added Pie figure which would depict the relative frequency in percentage of described genetic causes of acromegaly.

Include information on genetic panel availabilities for these diseases from clinical standpoint or laboratories which offer these genetic tests on clinical basis. Perhaps discuss estimation of cost.

Consider to divide the description to germline mutations and somatic mutations as separate entities 2 subchapters (not at the same level).  

Suggestions by lines:

Line 31:  The expression tumor grows implies growth hormone tumor growth, please describe the types of tumor growth of other organs reported in literature.. colon polyps etc.

 Line 36:  childhood and adolescents move to lines 33 prior to discussion of prevalence after the prevalence continue with line 38.

Line 39: the percentage of growth hormone tumors from all pituitary tumors shall be provided

Line 48: the percentage of sporadic versus the genetic causes reported in adulthood vs gigantism should be specified.

Line 49-56 :  This will be described in detail in the next paragraphs so these lines could be replaced by his  : Herein we provide overview of various genetic causes and circumstances where a genetic cause of acromegaly/gigantism have been described ( table 1) and the pie graph.

Line 60: the exact prevalence of FIPA is not known but appears to be more common than initially believed please list what was it initially believed at as this might confuse the reader.

Do you have current info on penetrance of FIPA (%)

Line 68 : the sentence is out of order on XLAG should be moved somewhere else 

Line 71: no need to described AIP positive prolactinoma family

Line 72 through 75 would be better to move after line 78 once the reader is familiarity with the  expression AIP

Line 132-134 is very interesting but too detailed detail for the audience

Line 139: Please explain the rational of genetic screening at age 4 however 1st MRI at age 10.

Line 97 and 147 is repetition in move some of the information from line 97 to 147. Reader might not be familiar what sparsely and densely granulated means, would explain it.

Line 207 appears to be a repetition of line 204.

Line 238 overview would move ahead of line 211 and include in the overview the definition of MEN4

Please be clear at which age MEN1 gene testing should be initially performed and which age biochemical testing and MRI should be initiated.

Line 236 and 237 can be removed as it is repetition of previous

Line 257 should be moved to line 253

Line 266 at the end please describe the percentage of patients with MEN 1 who have positive genetic mutation documented and move line 287 to this paragraph.  Line 282 through 286 could be removed.

Line 287 through 303 is not therapy but genetics of MEN4 and it can be shortened and moved to the 2nd part of genetics description (265).

Line 315 moved to 306

Line 306: Do have information how common is pituitary tumor in percentages versus hyperplasia in McCune Albright?

Diagnosis line 329 :shorten by removing line 329 and 330 diagnosis can be made clinically by complete physical and clinical evaluation of patients. Move the clinical features description to overview.

Line 359 paragraph can be moved ahead of line 349

Move the detailed description of clinical characteristics of Carney to overview and in the diagnosis line 384 just states 2 or more major criteria

 Line 410 and 411 should be moved genetics, in the genetics please state which types of SDH mutations have been associated with acromegaly was it the B,C, A or all?

I would recommend removing line 442-448.

NF 1  move the line 468 prior to 452,  I would recommend to shorten the detailed description of case  461 to 466. Please make a statement if clinicians need to test for NF1 gene or clinical diagnosis is sufficient.

I wonder if the other genes described line 533-549 are beyond the scope of this Journal interest.

Line 556 it would be useful to describe the existing gene panel testing which offers the tests

While the authors describe the recommended initial age for genetic testing, initial age for MRI and age for biochemical testing for many of the syndromes, please consider adding for all available.

Perhaps add in the table recommended age of screening initiation for offspring.

Reviewer 3 Report

Thank you for submitting this very comprehensive and informative review on genetic causes of acromegaly. 

Minor comments: 
- in your cohort of AIP / FIPA patients, could a prevalence be determined?
- in AIP-mutated individuals, is there an association with other subtypes of pituitary tumors?

- in presence of 2+ MEN-1 associated tumors, can authors discuss the added value of genetic testing compared to periodic calcium assessments ?

Reviewer 4 Report

The authors have read and reviewed a huge amount of literature on the Genetics of Acromegaly and Gigantism, and it was a great job!

However, in recent years, a number of papers have been published on the genetics of pituitary tumors, and these should not be ignored. Be sure to cite the relevant literature.

For example:

Endocrine. 2021 Feb 4. doi: 10.1007/s12020-021-02633-0

Endocrinol Metab Clin North Am. 2020 Sep;49(3):433-452.

Curr Opin Endocr Metab Res. 2018 Aug;1:19-24

Minor comments:

-L28, It would be better to describe how rare Acromegaly is. If necessary, you may cite recent papers such as Endocrinol Metab Clin North Am. 2020 Sep;49(3):347-355. and Acta Neurochir (Wien) 2020 Jun; 162 (6): 1317-1323.

-L30, It would be better to describe which joints are the most common.

-L55, The abbreviation for multiple endocrine neoplasia should be stated. Same as McCune-Albright Syndrome and Carney complex.

-L98, The space immediately after GH is unnatural.

-L139, Is there any reason to state "not later than 4 years of age" ? Please cite the literature if necessary.

-L210, Please write your textbook knowledge (outline, characteristics, etc.) about MEN4 anywhere in this section.

-L309, My understanding is that it is a common disease in the first decade.

-L325-326, "Arg201 or Gln227" may be kinder than "201 and 207" ?

-L401, It should be stated what PPGL stands for.

Round 2

Reviewer 1 Report

The authors have made extensive changes to the manuscript based on comments from the various Reviewers.  Its a very good review and the balance and completeness is very much improved.  The footnote on PitNets is particularly diplomatic.

There are a few new minor errors that have crept in

Line 84 needs to be rephrased (it is not clear what is being said)  There is impossible to establish the penetrance of AIPnegative FIPA. Only thing we know that it is lower than in AIPpositive FIPA as family members affected/family is smaller.

Line 148 "have been tried" instead of: experimental approaches has been tried to support clinical genetic decisions regarding

Line 500: if the or a patient instead of: The diagnosis 0can be made clinically if patient presents two or more signs of the condition.

Conclusion and figure: the gene for FIHP-familial hyperparathyroidism jaw tumour syndrome is mentioned.  Is there a role for this in the pituitary?  If so, then it should appear somewhere in the text possibly?

Reviewer 4 Report

The pointed out part has been corrected appropriately.

Author Response

The pointed out part has been corrected appropriately.

We thank the reviewer for his positive remarks.